# Correlation among Composition, Microstructure and Hardness of 7xxx Aluminum Alloy Using Original Statistical Spatial-Mapping Method

**DOI:** 10.3390/ma15165767

**Published:** 2022-08-21

**Authors:** Bing Han, Dandan Sun, Weihao Wan, Caichang Dong, Dongling Li, Lei Zhao, Haizhou Wang

**Affiliations:** 1Qingdao NCS Testing & Corrosion Protection Technology Co., Ltd., Qingdao 266071, China; 2Beijing Key Laboratory of Metal Materials Characterization, Central Iron & Steel Research Institute, Beijing 100081, China

**Keywords:** 7B05 aluminum alloy, deep learning, element distribution, recrystallization, spatial-mapping, data mining

## Abstract

The quantitative study of the relationship between material composition, microstructure and properties is of great importance for the improvement in material properties. In this study, the continuous data of elemental composition, recrystallization, hardness and undissolved phase distribution of the same sample in the range of 60 to 150 square millimeters were obtained by high-throughput testing instrument. The distribution characteristics and rules of a single data set were analyzed. In addition, each data set was divided into micro-areas according to the corresponding relationship of location, and the mapping between multi-source heterogeneous micro-area data sets was established to analyze and quantify the correlation between material composition, structure and hardness. The conclusions are as follows: (1) the average size of the insoluble phase in the middle of the two materials is larger than that of the surface, but due to the existence of central segregation, the average area of the T4 insoluble phase showed an abnormal decrease; (2) there was positive micro-segregation of Al, Cr, Ti, and Zr elements, and negative micro-segregation of Zn, Cu, and Fe elements in the recrystallized grains of the T5 middle segregation zone; (3) the growth process of the insoluble phase was synchronous with the recrystallization proportion and the size of the recrystallized grains; (4) the composition segregation and recrystallized coarse grains were the main reasons for the formation of low hardness zone in T4 and T5 materials, respectively.

## 1. Introduction

7B05 aluminum alloy has the characteristics of low density, corrosion resistance, weldability and thermal deformation. It has been widely used in high-speed train components such as traction beams, corbel beams and buffer beams [1,2,3]. There are obvious differences in mechanical properties of hot-rolled thick plates of high-strength aluminum alloy on the same plane and different planes along the thickness direction. This anisotropy mainly results from the deformation and uneven temperature distribution during material processing and heat treatment [4,5].

The mechanical properties of the rolled sheet along the thickness direction are different due to the differences in the surface and center structure [6,7,8,9,10]. During the rolling process, the surface of the plate comes into contact with the roller, resulting in an unstuck layer. A greater driving force is caused by the interaction between the dynamic recrystallization to produce full recrystallization organization [11]. Defects near the center of the point, such as dislocation density, are small, and the metal flow is relatively stable. Due to the high temperature, the dynamic response function is stronger, and the proportion of dynamic recrystallization to the surface is relatively low. However, due to the high temperature and long holding time in the central part, the deformed fiber microstructures can easily fuse with each other and form coarse recrystallized grains [12].

The composition difference between the surface and the central part is also an important reason for the difference of mechanical properties. The results show that 7B05 aluminum alloy has both microscopic segregation in several grain ranges and macroscopic segregation at the centimeter level [13]. Macroscopic segregation is due to the presence of element aggregation or impoverishment in the post-solidification center at high temperature during casting solidification. The results show that the center segregation of the billet can be inherited by the middle part of the plate after rolling [14]. The micro-segregation of 7B05 aluminum alloy shows a zonal distribution of elements corresponding to the distribution characteristics of the microstructure [15].

At present, the problems of low stability and poor durability seriously restrict the localization process of aluminum alloy materials for high-speed rail. The fundamental reason for this is that the internal micro-scale composition and structure of the material have low control accuracy and experience large fluctuations. Therefore, it is of great significance to study the uniformity of aluminum alloys in terms of the microscopic composition and structure. However, the existing characterization methods of composition and structure cannot achieve continuous data acquisition, and can only obtain local information to replace the overall state of the material, ignoring the influence of the inhomogeneity of the material itself. In this study, the concepts of material genetic engineering and big data [16,17,18,19,20,21] were adopted, and a high-throughput method was selected to characterize the structure–composition–property distribution and obtain the information of the continuous distribution of samples. At the same time, an original statistical spatial mapping method is proposed to establish the correlation between heterogeneous data.

The commonly used quantitative methods of organizational structure are usually implemented by image processing software along with manual participation. This process requires significant time and labor inputs; thus, the statistical results are acquired only from limited features and regions [22,23,24]. In view of the above shortcomings, in this study, image acquisition instruments were combined with computer vision methods, and a high throughput method of quantitative identification and statistical analysis of the aluminum alloy microstructure was devised based on deep learning [25,26]. After verification, this method was found to be accurate and comprehensive for the statistics of massive organizational data; please refer to our published articles, for the specific research process and results [27].

The common analysis methods for component distribution, such as scanning electron microscopy, energy spectrum analysis and electron probe microanalysis, are only aimed at micro-area test, with slow analysis speed and low quantitative sensitivity [28,29,30]. Spark spectrometry and ICP are discontinuous measurement techniques, so the global content distribution cannot be obtained [31,32]. Original position statistic distribution analysis (OPA) via spark source and via laser-induced breakdown spectroscopy (LIBS-OPA) can realize continuous distribution analysis in a large range, but creates the problem of surface damage. In contrast, microbeam X-ray fluorescence spectrometry has the advantages of high micro-area resolution, fast analysis speed, lack of surface damage and sustainable testing [33,34,35]. It has been widely used in the fields of material, archaeology and geological research.

Original statistical spatial-mapping technology is characterized by the originality, in-situ properties and statistics of information, which can reflect large-scale statistical distribution trends of the composition and microstructure of metal materials [36,37,38,39]. In this study, the deep learning method was used to quantitatively characterize the microstructure characteristics of four aluminum alloy pillow beams. The element distribution was characterized by microbeam X-ray fluorescence spectrometry. Using the idea of original statistical spatial-mapping characterization, the correlations among composition, hardness and microstructure from multi-source heterogeneous data sets were analyzed.

## 2. Materials and Method

### 2.1. Experimental Samples

The 7B05 aluminum alloy of the same brand and four specifications for commercial sleeper beams was used in the experiment. This high-speed rail material was provided by China Railway Rolling Stock Corporation (CRRC, Qingdao, China). After hot rolling, the material was heat treated with natural aging (T4) and artificial aging (T5). Its chemical composition is shown in Table 1. Samples were cut from the rolled aluminum plate with a size of 10 mm × 10 mm × 6 mm, 10 mm × 10 mm ×10 mm and 10 mm × 10 mm × 15 mm (thickness), and the sampling surface remained perpendicular to the rolling direction. Depending on heat treatment and sheet thickness, these four aluminum alloys were named T4-6, T4-15, T5-10, and T5-15.

### 2.2. Quantitative Method of Insoluble Phases Using High Throughput Scanning Electron Microscope

The polished sample was placed in the Navigator OPA high throughput scanning (NCS Testing & Technology Co., Ltd., Beijing, China) electron microscope to collect images. The acquisition field sizes were as follows: T4-6 aluminum alloy, 10 mm × 6 mm; T4-15 aluminum alloy, 10 mm × 15 mm; T5-10 aluminum alloy, 10 mm × 10 mm; T5-15 aluminum alloy, 10 mm × 15 mm. The number of images collected from each of the four samples was 3362 (T4-6), 11,508 (T4-15), 7056 (T5-10) and 10,668 (T5-15), and the pixel size of each image was 4096 × 4096. The magnification of T4-15, T5-10 and T5-15 images was 2800 times, and of the T4-6 image was 2000 times.

The quantitative statistical characterization method of the second phase was established based on the deep learning algorithm. The experimental process is shown in Figure 1. The main steps are as follows: building the image recognition framework (U-Net) based on an artificial neural network [40,41,42], making the training set, training the feature data set to obtain the U-Net segmentation model, inputting the sequence test images obtained by high-throughput scanning electron microscopy into the trained model, extracting the image features to be tested, and evaluating the randomly selected test results through common segmentation indexes to ensure the accuracy of the quantitative method.

The network we constructed in the dotted box based on the original U-Net architecture consisted of a contracting path and an expanding path [43,44,45]. The segmented image data set was processed by the connected-region algorithm, and the feature distribution data set was characterized by the mathematical statistical method. The location information of extracted features was restored to the aluminum alloy section, and the quantitative statistical results with spatial distribution information could be obtained. For verification of the method accuracy and results, refer to the previous results published by our research group [46].

### 2.3. Quantitative Method of Gray Value Using Optical Microscopy

Firstly, samples were polished with 400#, 800#, 1200#, 2500# and 4000# sandpaper, and then polished with 0.5 µm silicon dioxide and a fine polishing cloth until the surface scratches were completely removed. Secondly, samples were eroded with the erosion solution “Keller reagent”, and the erosion time was about 40 s. Microstructure images were acquired using the Leica DM6000 metallurgical (Leica Microsystems CMS GMBH, Wetzlar, Germany) microscope with autofocus and continuous acquisition mode. The field sizes were as follows: T4-6 aluminum alloy, 10 mm × 6 mm; T4-15 aluminum alloy, 10 mm × 15 mm; T5-10 aluminum alloy, 10 mm × 10 mm; T5-15 aluminum alloy, 10 mm × 15 mm. The magnification was 50 times, the number of images collected from the four samples was 24 (T4-6), 50 (T4-15), 35 (T5-10) and 50 (T5-15), and the pixel size of each image was 2400 × 1800. Then, the same parameters were set to stitch images so that a full-field image could be obtained for one sample. Finally, the stitched image was processed to increase the contrast. According to the threshold of micro-area division, the image matrix was divided by MATLAB software (Matlab R2021a, Mathworks, Nedick, MA, USA.), and the average gray value of each micro-area was obtained.

### 2.4. Instruments and Conditions of Composition Test

The element composition distribution of the aluminum alloy section was analyzed by microbeam X-ray fluorescence spectrometer (M4 tornado, Bruker, Karlsruhe, Germany). The detailed parameters were as follows: X-ray tube voltage was 50 kV, current was 150 μA, target material was Rh, beam spot size was 20 μm, beam spot collection interval was 10 μm, scanning time per pixel was 100 ms, and sample chamber vacuum was 20.1 mbar.

### 2.5. Instruments and Conditions of Hardness Test

According to the ISO 6507-1-2005 standard, a Vickers hardness tester (Qness Q10, Qness GmbH, Salzburg, Austria) was used to measure the hardness of the aluminum alloy section. The multi-point measurement mode was adopted to ensure that the distance between the centers of two hardness points was greater than 3 times the diagonal distance. The test load was 10 N and the loading time was 10 s. The hardness matrix data with equal intervals on the sample surface was processed by interpolation, and the interpolation matrix could be considered to represent the actual hardness distribution of the sample cross section. Taking T5-10 as an example, the measurement area was 10 mm × 10 mm, and the number of hardness points was 4150.

### 2.6. Instruments and Conditions of Electron Back Scattered Diffraction (EBSD)

After cutting, the sample was polished with 400#, 800#, 1200#, 2500# and 4000# sandpaper, and then polished with aluminum oxide and a velvet cloth until the surface scratches were completely removed. In order to further eliminate the residual stress on the surface of the sample, electrolytic polishing was carried out. The polishing liquid was 25% H_3_PO_4_ + 25% C_2_H_5_OH + 50% H_2_O mixed solution. The electrolysis voltage was 15 V, and the electrolysis time was 30 s. EBSD was observed by a TESCAN S8000G (Tescan GmbH, Brno, Czech Republic) scanning electron microscope; the scanning working distance was 14 mm, the acceleration voltage was 20 KV, the current was 3 nA, the step size was 0.5 μm, and the scanning area size was 500 pixels.

## 3. Experimental Results

### 3.1. Microscopic Morphology and Characteristics of Aluminum Alloy

#### 3.1.1. Morphology and Characteristics of Crystallography

The surface layer structure is composed of equiaxed crystals and abnormal grains grown after recrystallization, as shown in Figure 2a,e,i,m, because more deformation energy stored in the surface during rolling and heat treatment leads to re-nucleation and growth of grains. As shown in Figure 2b,f,g,n, the center grains are significantly elongated along the rolling direction, showing obvious fiber characteristics. The grain size is not completely uniform, and there is a certain level of mixed crystal phenomenon. Figure 2 shows that the recrystallization degree of the thick plate is limited, and most areas are still dominated by deformed sub crystals.

It can be seen from IPF that the maximum orientation densities of the surface and the center of the two materials T4-6 and T5-10 are quite different. Because of the thickness of the sheet, T4-6 and T5-10 are more significantly affected by hot rolling deformation. Figure 2a–d shows that the orientation of T4-6 is the transition direction from <001> to <111>, and the orientation density of the center is greater than that of the surface layer. Figure 2i–l shows that the main orientation of the surface is relatively dispersed, and its distribution is the transition direction of <101> to <111>, the transition direction of <001> to <101> and the transition direction of the <111> direction, and the middle layer is concentrated in the <101> direction. T4-15 and T5-15 alloys have more dispersed surface and intermediate microstructure orientations. T4-15 is mainly distributed in the transition direction, whereas T5-15 is mainly concentrated in <101> and <111> orientations.

As is conveyed by Figure 3b,e,h,k and Figure 3c,f,i,l, the white area in the middle layer is generated from the growth in recrystallized grains, which are resistant to corrosion due to their low internal crystal defects and dislocation density. The gray-black area in the middle layer has fine grains and is composed of deformed sub-grains. Due to crystal defects and dislocation density, a large number of sub-grain boundaries are formed, which are easily corroded, so that the structure appears black. The surface structure is also gray-black after corrosion, and its gray value is much smaller than that of the sub-grains in the middle layer. The white area is mainly composed of large grains and shows similar topological morphology to that of the matrix. After corrosion, the white area is smooth and the grain boundary is irregular. The gray-black area has fine grains, and the surface is uneven after corrosion. The grain boundaries in the gray-black area are not fully revealed, and some grains still maintain the slender crystal state.

The dislocation angle distribution and grain size of the surface and center layers of the 7B05 material in the range of 500 μm × 500 μm are shown in Figure 4. As can be seen from Figure 4a,b, a small angle grain boundary is dominant in both the surface layer and central layer. When the grain boundary is larger than 15 °, the proportion of the large angle grain boundary decreases rapidly. The proportion of the large angle grain boundary in the surface layer is larger than that in the central layer. Figure 4a,b shows that the degree of recrystallization of the surface layer is greater than that of the middle layer. As can be seen from Figure 4c,d, when the grain diameter is greater than 4.56 μm, the proportion of grains decreases rapidly. The proportion of grains larger than 4.56 μm in T4 material is 0.08~0.45, while the proportion in T5 material is 0.2~0.63. The number of large grains formed by recrystallization of T5 material is more than that of T4. When the surface grain size of T4-15 is greater than 20.56 μm, there is a sudden-change point, which is related to the abnormally grown structure formed by surface recrystallization. The corresponding dislocation angle of T4-15-E becomes larger in the range of 35°–50°, as shown in Figure 4a.

#### 3.1.2. Morphology and Characteristics of Insoluble Phase

Figure 5 shows the morphology of undissolved and coarse second phases in the backscattering mode. The refractory phases distributed on the matrix are mainly white and gray. The sizes of the inclusion phases are different, their shapes are an irregular polygon, and their distribution at different positions of the rolled plate section is not uniform. Since the Cu content in 7B05 is small, only 0.022–0.23%, as shown in Table 1, the white inclusion components may exist in the form of Fe and Si compounds, such as AlFeMnCr, AlFeMnCrSi, and Al_7_Cu_2_Fe. The black inclusion phases have high content of Mg and Si and may exist in the form of compound Mg_2_Si.

### 3.2. Statistical Results of Second Phase Distribution

The U-Net target model based on deep learning was used to segment the image to be measured. After segmentation, the full field (60~150 mm^2^) distribution of the second phase on the sections of four aluminum alloy rolled plates was counted. Figure 6 shows the total insoluble phase number, area and average size per 512 pixels along the thickness direction. It can be seen from the figures that there are insoluble phases at different thickness positions, and the area, number and density of the second phase show different distribution trends with the change in thickness.

For T4 materials experiencing natural aging, the total area, quantity and average area have a basically symmetrical distribution. The area of insoluble phases from surface to heart decreased slightly; the number of insoluble phases decreased significantly from the surface to the heart, but increased near the central layer; and the change trends of the average area in these two T4 materials are the same, rising from the surface layer at 1.5 and 1 μm^2^, to the center 3 μm^2^. Two higher peaks appear near the central layer, then extend to the central layer, and there is a “valley” with decreasing average area.

The total area, quantity and average area of T5 artificial aging materials are asymmetrically distributed. The area change trend of the insoluble phase from the surface layer to the central layer is more significant compared with T4. The amount of insoluble phase decreases from the surface layer to the central layer. The average area of the insoluble phase increases from surface to center. During the hot forming process of the rolled sheet, the surface layer and the middle layer are subjected to different forces and different holding times. As a result, the deformation degree of different parts of the rolled sheet is different, and the degree of fragmentation of the intermetallic compound is different. In addition, the holding time and element segregation will also affect the quantity and area of the insoluble phase.

### 3.3. Statistical Results of Element Distribution

The element mapping of each position was performed in the 10 mm × 15 mm section of T4-15 aluminum alloy by means of microbeam X-ray fluorescence, and the composition distribution is shown in Figure 7. The results show that there is a segregation zone in the central layer region, where Al, Cr, and Zr elements show positive segregation, and Fe and Zn elements show negative segregation. The content of Fe element in the segregation area is low and the distribution is uneven, the distribution is relatively uniform in the area near the surfaces, and the distribution on the cross-section is point-like aggregation. There is no obvious segregation band in the central layer of Mn element, but there is a strip-like aggregation distribution with high content.

As can be seen from Figure 8, the contents of Al, Ti, Zr, Zn and Cu elements in the surface layer have a small variation range and relatively uniform distribution. The low-content and high-content elements in the segregation area are distributed alternately, and the distribution state is band-shaped. Based on the scanning spot of 20 μm, the scanning distance was set to 10 μm, and the crystal grain size was in the micron level; therefore, it is judged that the micro-segregation morphology is similar to the distribution of the structure. When the alloy sheet is calendered, the distribution of stress and temperature on the surface and center layers are not uniform, and the surface in contact with the roller cools fast, causing the alloy elements to be strongly dissolved in the upper and lower layers of the sheet. Therefore, the surface layer has a high content of alloy elements that are solid-dissolved in the matrix and are evenly distributed. Extending from the surface layer to the inside, the cooling rate of the rolled plate slows, so that the solute redistribution of the alloying elements occurs, and the elements of the central layer continue to diffuse during the deformation process. In the central part where the temperature is high and the holding time is long, the contents of Ti, Zr, Zn, Cu and other alloying elements dissolved in the matrix decrease, and the elements exist in the form of precipitation, so that the contents of elements in the segregation zone vary greatly.

The precipitation characteristics of Fe element are different from those of other elements, and are mainly related to the impurity phase. The distribution of Fe exhibits point aggregation, and surfaces contain evenly distributed Fe with high content. Due to the high temperature and long holding time of the interlayer, the content of solid solution in the matrix decreases, and the second phase is fully precipitated and easy to grow. The content of Fe in the precipitated phase differs greatly from that in the matrix.

### 3.4. Statistical Results of Hardness Distribution

The distribution trend of 7B05 section hardness is shown in Figure 9, and its characterization section is the same as the interface used in the above element distribution and microstructure characterization. Figure 9a,b shows that the hardness value of T4 section is in the range of 120–140 HV, the central layer has a low hardness area, and the hardness value below 125 is basically distributed in this area. As shown in Figure 9c,d, the hardness value of the entire T5 section is in the range of 90–120 HV, and the distribution along the thickness direction is not symmetrical. Taking T5-10 as an example, the stable thickness is close to 2 mm, whereas, on the other side, it is about 0.5 mm. In addition, the hardness distribution is also not uniform: the surface hardness is low, and the transition zone has high hardness and large fluctuations. There are band-shaped low-hardness zones on the middle layer, and the values below 100 are basically distributed in this area.

## 4. Statistical Mapping Trend Analysis

### 4.1. Analysis of Insoluble Phase-Structure Statistical Mapping Trend

As mentioned in Section 3.1.1, the gray value of the material under the light microscope is determined by the structure. The recrystallized organization is white after corrosion and has a large gray value; the deformed sub-grain is gray-black after corrosion and has a small gray value; the surface recrystallized organization is black after corrosion and has the smallest gray value. The morphology and grayscale of four aluminum alloys sections after corrosion are shown in Figure 10. However, the grayscale change in T4-15 after etching is opposite to that of other materials, which is presumed to be related to the abnormally coarse recrystallized structure in the surface layer, as shown in Figure 2e and Figure 4a,c.

The abscissa in Figure 11 is the number of images divided along the thickness direction, with 512 pixels as the threshold, representing the thickness of the rolled plate; the ordinates are the average area of the insoluble phase and the average gray value corresponding to each row of images. The distribution of the average area of the insoluble phase in the T4 and T5 materials along the thickness direction is related to the average grayscale change of the samples, showing a trend of a low surface layer and a high middle layer; see Figure 11. The aluminum plate in contact with the roller is subjected to forces such as rolling pressure, shearing force, and tension during rolling. With the increase in deformation degree, brittle intermetallic compounds tend to break into small and continuous undissolved phases. Due to the frictional force, the surface layer bears greater shear stress, and deforms more than the middle layer. Therefore, the T4 and T5 materials have more insoluble phases on the surface layer. When the surface is extremely cold, the precipitated insoluble phase has no time to grow. The middle layer has a long holding time, and the insoluble phase is easy to grow. However, T4 shows a low point in average area near the center layer. Combining Figure 6 and Figure 7, Fe element starvation and Zr and Ti element enrichment occur at this position, and there is a peak in the number of the second phase. We speculate that the number and size of the insoluble phase in the central layer is changed due to the segregation of solute elements.

The number of insoluble phases is also low in the position (image sequence number 200, 400, 600) where the gray value drops sharply based on Figure 11c. The average gray varies in a manner consistent with the average area of the insoluble phase, with a distribution trend of increasing–maintaining–decreasing along the thickness direction. Since the gray value is determined by the size and proportion of the recrystallized grains, the growth process of the insoluble phase is synchronized with the changes in recrystallization ratio and grain size. It is difficult to characterize the wide-ranging recrystallized grain size and the grain change process, whereas it is relatively easy to characterize the change trend of the insoluble phase. Therefore, this conclusion can be used to judge the change in the recrystallized grain size.

### 4.2. Analysis of Structure-Composition Statistical Mapping Trend

It is difficult to use a point-to-point method to explore the correlation between the two sets, since the component and grayscale data are two sets of data having different scales and different types. This study used the micro-region–micro-region method to explore the data rules and determine the degree of correlation between the two sets. The T5-10 section (10 mm × 10 mm) was divided into a 14 × 14 micro region, and the element content and grayscale data of 186 areas at the corresponding positions were extracted. The relationship between grayscale and element content is shown in Figure 12. Cr, Ti and Zr have similar data distributions, and Zn and Cu have similar data distributions.

It can be seen from Figure 12 that the areas with high average gray values have high contents of Al and Cr, indicating that the contents of Al, Cr, Ti and Zr in the recrystallized grains are high after forming and heat treatment, and these elements exist in the form of microscopic positive segregation. The areas with high average gray values have low Zn and Cu contents, indicating that the recrystallized grains have low Zn and Cu element contents, and these elements exist in the form of microscopic negative segregation. The wide range of micro-segregation of aluminum alloy seriously affects the morphology of the structure and the comprehensive performance. This conclusion is important for characterizing the degree of recrystallization segregation of aluminum alloy.

With the increase in gray value, the contents of Cr, Ti and Zr in the front part decrease first and then increase. The locations with the lowest gray values are distributed near the surface, and the degree of solid solution is the highest in this area. In the process of centripetal extension, the decrease in the content caused by the weakening of the solid solution is greater than the increase in the content caused by the proportion of recrystallization. Therefore, the front part of the fitting lines of Cr, Ti and Zr elements shows a decreasing trend. However, there is no change in the solid solubility of the Al element, and the distribution shows a slowly increasing trend. The statistical mapping trend of T4 is similar to that of T5, as shown in Figure 13.

### 4.3. Analysis of Structure-Hardness Statistical Mapping Trend

The section was divided into micro-areas for mapping, the gray value and hardness data of the corresponding micro-areas were extracted, and the abnormal data having a value of less than one tenth was filtered out. The relationship between gray value and hardness is shown in Figure 14. Figure 14a shows that the hardness value and gray value of T4-6 change in the same way, and the gray and hardness are both low on the surface layer and the near-intermediate layer. Figure 14b shows that the hardness value and gray value of T4-6 have opposite trends. The above results prove that the formation of the central low hardness is not determined by the gray value. The composition segregation near the central layer is significant (Figure 7), and the Zn element, which plays the main strengthening role, exhibits negative segregation here [44]. Therefore, the low-hardness zone in T4 is mainly caused by the negative segregation of alloy elements.

Figure 14c,d shows that, for the cross section of T5 material, the surface layer of the rolled sheet with the lowest gray value and low hardness consists of equiaxed grains formed by recrystallization. As the gray value increases, the hardness reaches the maximum. At this time, the structure is composed of a certain proportion of recrystallized grains and fine sub-crystals, and the grains are significantly elongated along the rolling direction, showing obvious fiber characteristics. When the gray value continues to increase, the hardness begins to decrease, and the position with a band-like distribution and the smallest hardness appears inside the section. The structure here is dominated by the deformed coarse recrystallization, and it is also the region where the component segregation is the most significant.

In summary, the hardness distribution is related to the structure. The surface equiaxed crystal has a small hardness value and uniform distribution; with the occurrence of T5 material recrystallization, the hardness in the transition zone rises to the highest value; when the recrystallization ratio is the highest, the hardness is the lowest. The recrystallized grain size on the intermediate layer of T4 material is smaller than that of T5 (Figure 2 and Figure 4), and there is no low-hardness zone with a band-like distribution. The low-hardness zone is mainly caused by the negative segregation of alloy elements.

### 4.4. Correlation among Composition, Hardness and Microstructure Using Original Statistical Spatial-Mapping Method

The original statistical mapping technique used in this study is based on the identification of intrinsic heterogeneity to characterize aluminum alloy samples with high-throughput and cross-scale statistical distribution, resolving the correlation building problem among element distribution, properties and microstructure over a large area. The region–region original statistical mapping method was selected, and the original data set was divided by selecting an appropriate region as the minimum unit. The characterization parameters of the micro-area in the muti-source data set were extracted; this process is a nonlinear down-sampling function, which reduces the space size in exchange for the increase in the dimension of parallel representation data. Through the mining of multi-source heterogeneous data sets, the correlation model between each data set was finally established, as shown in Figure 15.

## 5. Conclusions

(1)The number of insoluble phases in T4 and T5 has the same variation trend, with more in the surface layer and fewer in the middle. The average area of the insoluble phase in two T4 materials increases from 1.5 and 1 μm^2^ in the surface layer, to 3 μm^2^ in the center layer. The anomalous decrease in the insoluble phase area near the center is related to the center segregation. The growth in the insoluble phase of T5 materials is synchronized with the growth process of recrystallization.(2)The surface of the rolled sheet has a high degree of solid solution, and the composition is evenly distributed; for T5 material, there is positive segregation of Al, Cr, Ti and Zr, and negative segregation of Zn, Cu and Fe in the recrystallized grains of the middle segregation zone; T4 material has genetic center segregation caused by casting and hot rolling process.(3)The hardness value of the surface composed of equiaxed and coarse recrystallized grains is small and the distribution is uniform; the band-like hardness distribution in the T5 intermediate layer is related to the recrystallization proportion and the size of the recrystallized grains. The coarse grain in the T4 intermediate layer is smaller than that of the T5 material; therefore, there is no low-hardness zone with a band-like distribution. The low hardness region with a value below 125 in T4 is mainly caused by the negative segregation of alloying elements. Values below 100 are basically distributed in this area.(4)The original statistical spatial-mapping method demonstrated the feasibility of analyzing the correlation among massive multi-source heterogeneous data, including for composition–microstructure–performance.

## Figures and Tables

**Figure 1 materials-15-05767-f001:**
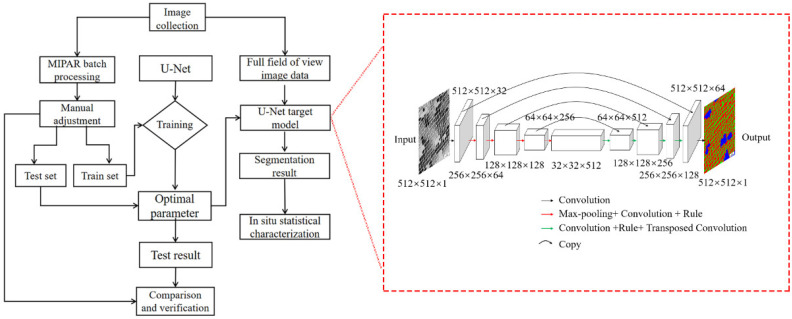
Quantitative characterization process of second phases and architecture of neural network employed in our method.

**Figure 2 materials-15-05767-f002:**
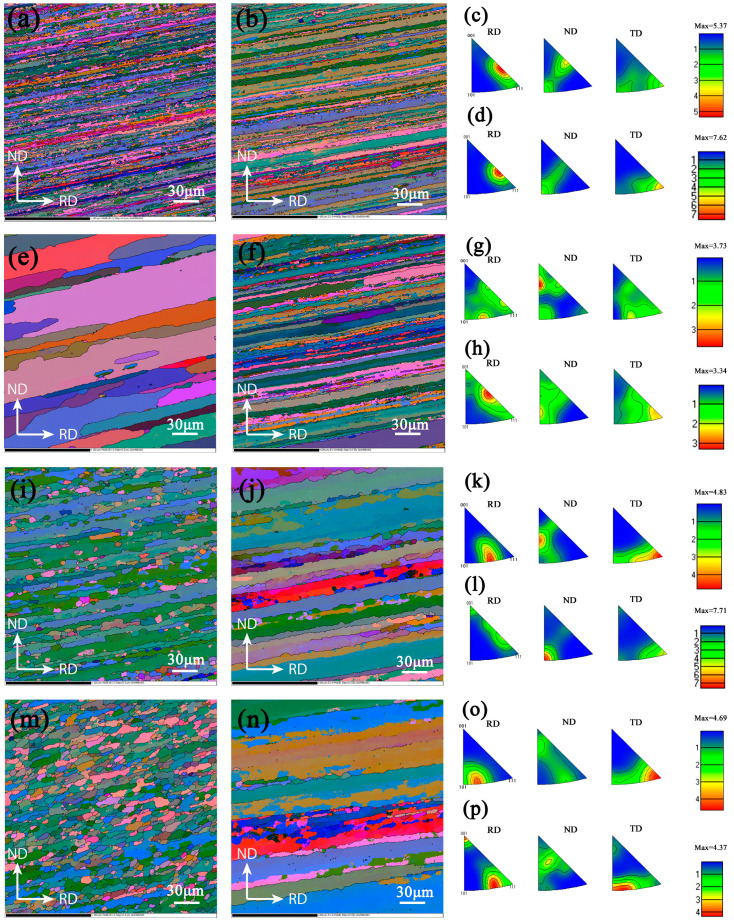
Orientation maps of 7B05 aluminum alloy based on EBSD measurements: (**a**) T4-6 surface microstructure, (**b**) T4-6 center microstructure, (**c**) T4-6 IPF of surface, (**d**) T4-6 IPF of center; (**e**–**h**) the corresponding T4-15 EBSD microstructures; (**i**–**l**) the corresponding T5-10 EBSD microstructures; (**m**–**p**) the corresponding T5-15 EBSD microstructures.

**Figure 3 materials-15-05767-f003:**
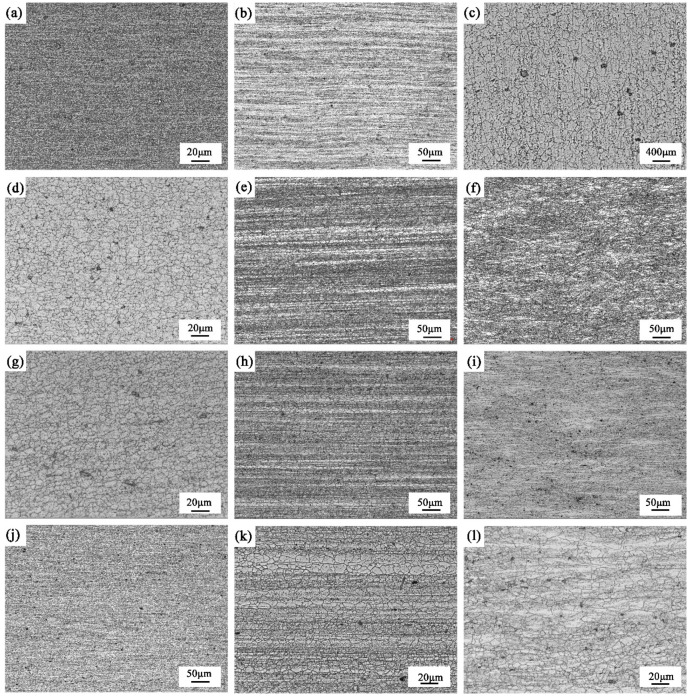
Micrographs of different sections in 7B05 aluminum alloy: (**a**) parallel rolling direction of T4-6, surface, (**b**) parallel rolling direction of T4-6, center, (**c**) vertical rolling direction of T4-6, center; (**d**–**f**) the corresponding T4-15 microstructures; (**g**–**i**) the corresponding T5-10 microstructures; (**j**–**l**) the corresponding T5-15 microstructures.

**Figure 4 materials-15-05767-f004:**
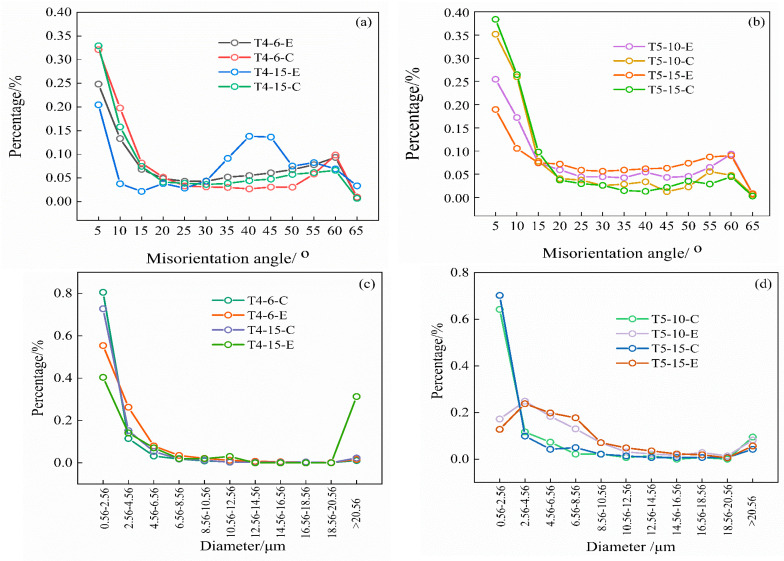
Distribution trend of misorientation and grain size of 7B05 aluminum alloy, where E represents the surface layer of the rolled plate, C represents the center layer of the rolled plate: (**a**) misorientation distribution of T4; (**b**) misorientation distribution of T5; (**c**) diameter distribution of T4; (**d**) diameter distribution of T5.

**Figure 5 materials-15-05767-f005:**
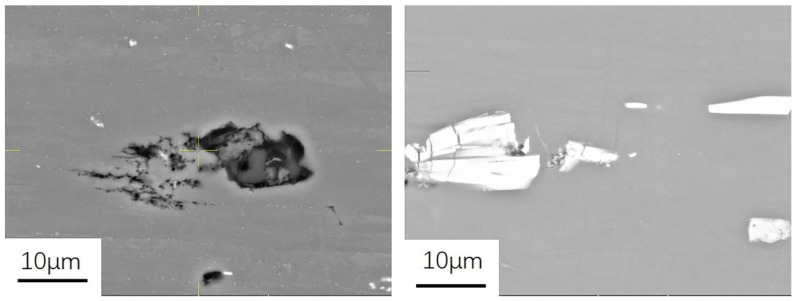
The morphology of undissolved second phases in 7B05 aluminum alloy.

**Figure 6 materials-15-05767-f006:**
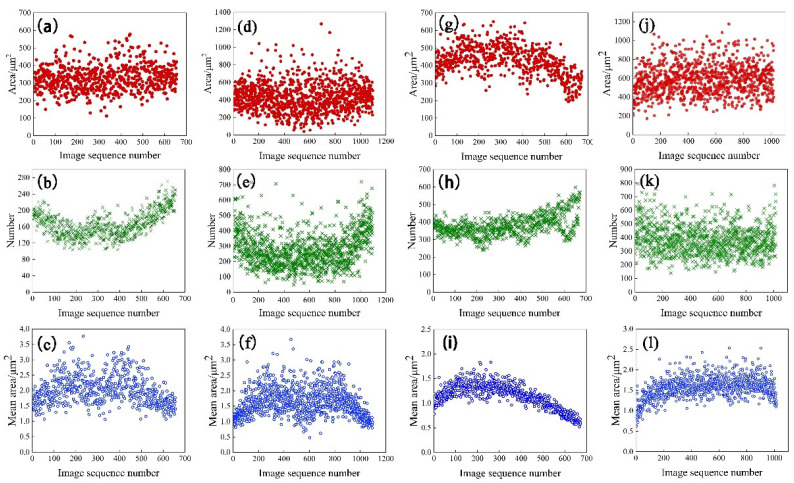
The total area, quantity and average area distribution of insoluble phases along the thickness direction in 7B05 aluminum alloy: (**a**–**c**) T4-6; (**d**–**f**) T4-15: (**g**–**i**) T5-10; (**j**–**l**) T5-15.

**Figure 7 materials-15-05767-f007:**
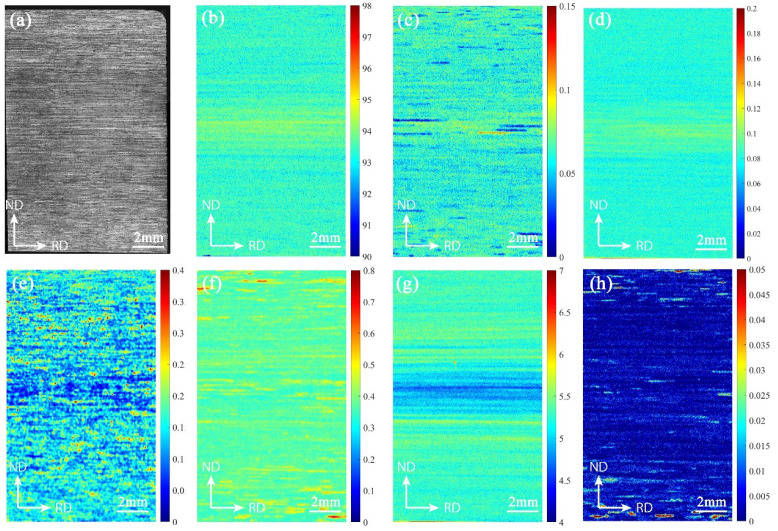
The mapping of T4-15 element contents measured by X-ray fluorescence: (**a**) sample; (**b**) Al element; (**c**) Cr element; (**d**) Zr element; (**e**) Fe element; (**f**) Mn element; (**g**) Zn element; (**h**) Cu element.

**Figure 8 materials-15-05767-f008:**
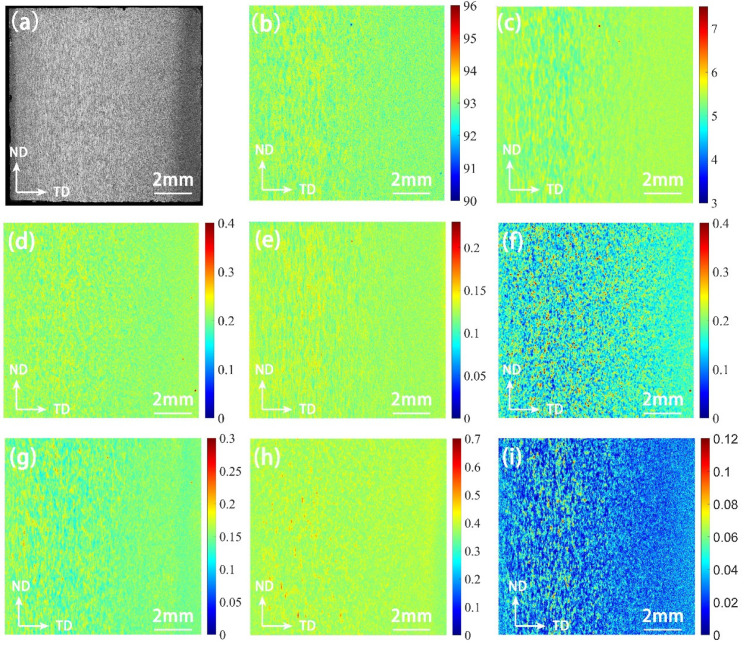
The mapping of T5-10 element contents measured by X-ray fluorescence: (**a**) sample; (**b**) Al element; (**c**) Zn element; (**d**) Cr element; (**e**) Zr element; (**f**) Fe element; (**g**) Cu element; (**h**) Mn element; (**i**) Ti element.

**Figure 9 materials-15-05767-f009:**
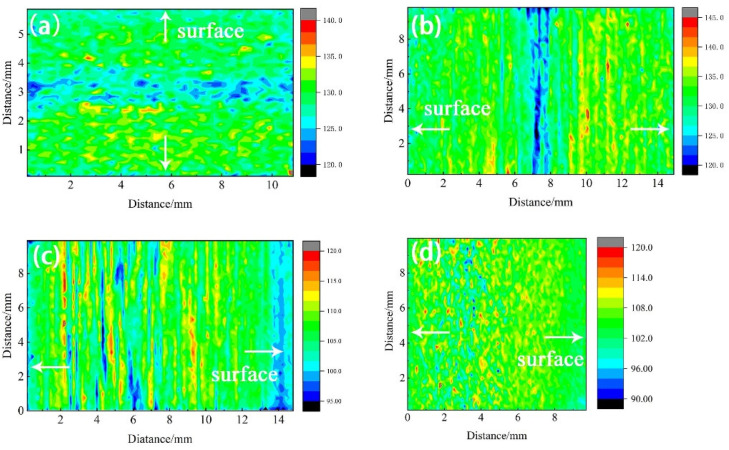
Hardness mapping of 7B05 aluminum alloy section: (**a**) T4-6; (**b**) T4-15; (**c**) T5-15; (**d**) T5-10.

**Figure 10 materials-15-05767-f010:**
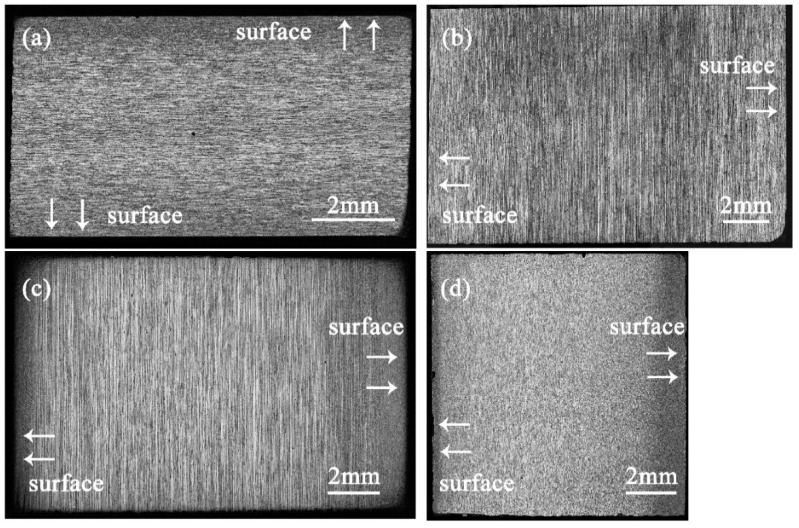
The surface morphology and grayscale of the four aluminum alloy sections after corrosion: (**a**) T4-6; (**b**) T4-15; (**c**) T5-15; (**d**) T5-10.

**Figure 11 materials-15-05767-f011:**
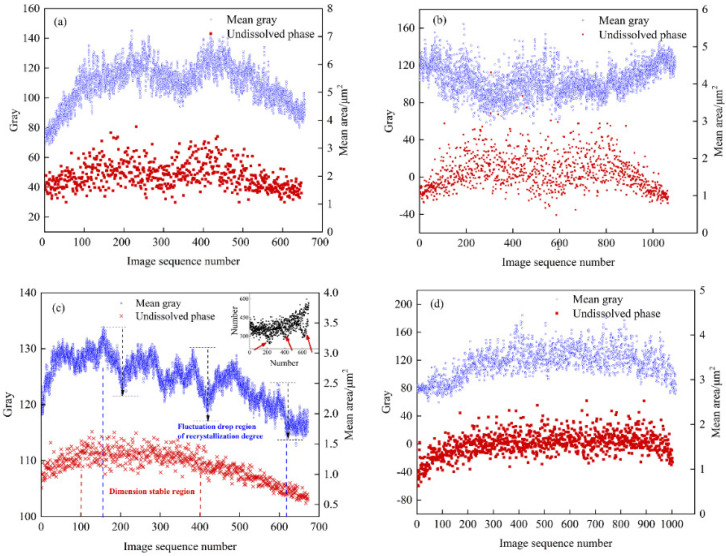
Distribution line diagram of gray and insoluble phases along the direction of section thickness in four aluminum alloys: (**a**) T4-6; (**b**) T4-15; (**c**) T5-10; (**d**) T5-15.

**Figure 12 materials-15-05767-f012:**
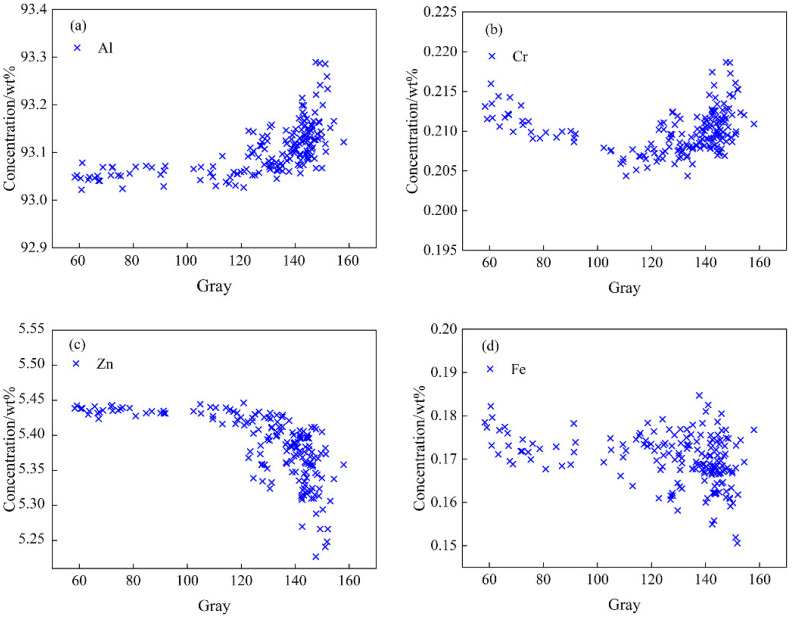
Statistical mapping diagram of gray and component in T5-10 aluminum alloy microregion. (**a**) Al element; (**b**) Cr element; (**c**) Zn element; (**d**) Fe element.

**Figure 13 materials-15-05767-f013:**
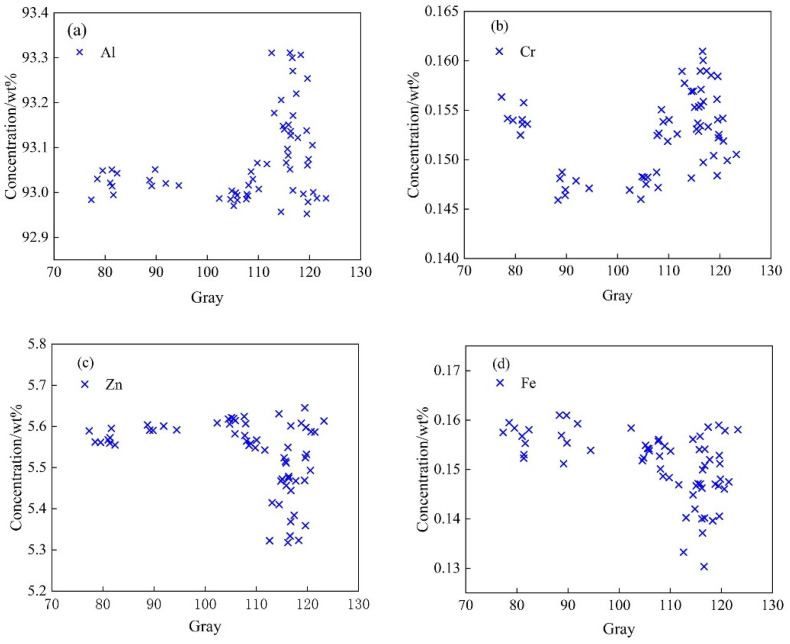
Statistical mapping diagram of grayscale and component data in the T4-6 aluminum alloy microregion. (**a**) Al element; (**b**) Cr element; (**c**) Zn element; (**d**) Fe element.

**Figure 14 materials-15-05767-f014:**
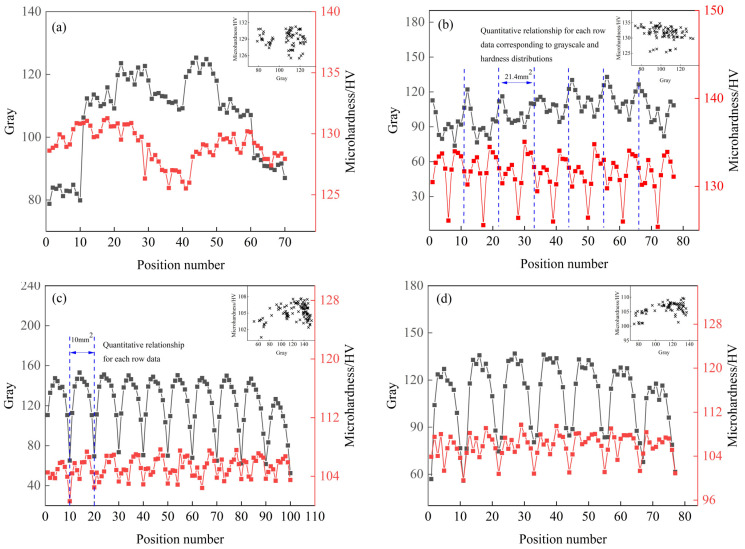
Statistical mapping diagram of gray and hardness data in four aluminum alloy microregion: (**a**) T4-6; (**b**) T4-15; (**c**) T5-10; (**d**) T5-15.

**Figure 15 materials-15-05767-f015:**
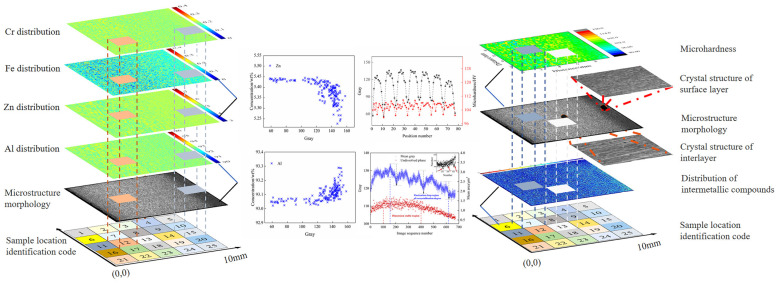
Spatial-mapping model of composition–structure–hardness in 7B05 aluminum alloy.

**Table 1 materials-15-05767-t001:** Chemical composition of four aluminum alloys (wt%).

Material	Zn	Mg	Cu	Fe	Si	Mn	Cr	Zr	Ti
T4-6	4.53	1.1	0.23	0.17	0.088	0.34	0.18	0.12	0.046
T4-15	4.39	1.38	0.022	0.16	0.067	0.35	0.084	0.071	0.02
T5-10	4.31	1.01	0.15	0.17	0.062	0.37	0.23	0.097	0.05
T5-15	4.23	1.09	0.16	0.17	0.058	0.37	0.22	0.11	0.048

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
