# Peer review of "Correlation among Composition, Microstructure and Hardness of 7xxx Aluminum Alloy Using Original Statistical Spatial-Mapping Method"

_materials, 2022, doi:10.3390/ma15165767_

Round 1
Reviewer 1 Report
Abstract part:
- line 10, after „Abstract“ which is in Bolt the word Parallel should not be Bolt,
- the whole abstract does not correlate with the paper title, and needs to be rewritten,
Introduction:
- Introduction, the title of the whole part is even missing,
- Introduction as itself needs to be rewritten, in the state as it is written it does not provide sufficient information about the issue addressed in the article,
Part Materials and method
- Please see the formatting of the title – line 75 and correct the chapter numbering according to the note above,
- What does CRRC abbreviation mean? – line 78. Please explain
- Your experimental material is 7B05 aluminium alloy; please explain the sentence – line 79 – “Samples were cut from the rolled steel plate with a size .......... „ ….. so what was your experimental material?
- Table 1 - line 85, please check the formatting, different types of font used in the table caption and in the body,
- What this sentence does mean? 3362, 11508, 7056, 10668 and 4096 continuous images with 4096×4096 pixels were collected respectively. – lines 90, 91
Part Experimental results
- Again, please keep formatting and fonts, once used bolt, then italic and even regular type …..,
- Lines 131 and 133, image numbering should be separated by commas……,
The whole article is written very chaotically. Different fonts are used, there are many mistakes in the text and it seems to me like a "quickly written" article. It lacks a comprehensive concept, it is very difficult to read, and the quality of the images and their descriptions are low quality and unreadable. For example pictures 11, 12, 13 and 14 description of the axis "x" or "y" - what does Gray mean?
In the "Conclusion" section, specific values are missing, e.g. a number of insoluble phases are mentioned in the discussion, but the results of the experimental part should be summarized in the conclusion.
My recommendation is to revise the entire article in a significant way and to strictly follow the formatting so that it is uniform throughout the article. In terms of content, the article is fine, but it lacks a concept. My recommendation - significant "Major revision"
Author Response
- line 10, after „Abstract“ which is in Bolt the word Parallel should not be Bolt,
The font and format of abstracts have been unified.
- The whole abstract does not correlate with the paper title, and needs to be rewritten,
The abstract has been rewritten.
Introduction:
- Introduction, the title of the whole part is even missing,
The second, third and fourth paragraphs have been added to the introduction section, summarizing the correlation between compositional structure and properties in 7-series aluminum alloys.
- Introduction as itself needs to be rewritten, in the state as it is written it does not provide sufficient information about the issue addressed in the article,
The introduction section has been rewritten. The introduction clarifies the significance of the research on the correlation of 7 series aluminum alloys. In addition, the influence of production process on composition, structure and properties is clarified. The problems existing in the current research methods of composition and structure correlation are pointed out. A research method to establish correlation using in situ statistical mapping method combined with high-throughput data acquisition equipment is proposed.
- Please see the formatting of the title – line 75 and correct the chapter numbering according to the note above,
The chapter numbering has been corrected.
- What does CRRC abbreviation mean? – line 78. Please explain
The full name of CRRC has been added. – line 102
- Your experimental material is 7B05 aluminium alloy; please explain the sentence – line 79 – “Samples were cut from the rolled steel plate with a size .......... „ ….. so what was your experimental material?
The steel plate has been modified to aluminum plate. – line 104
- Table 1 - line 85, please check the formatting, different types of font used in the table caption and in the body,
The formatting and types of font used in the table caption and in the body has been unified.
- What this sentence does mean? 3362, 11508, 7056, 10668 and 4096 continuous images with 4096×4096 pixels were collected respectively. – lines 90, 91
The number of images collected from four samples are 3362 (T4-6), 11508 (T4-15), 7056 (T5-10), 10668 (T5-15), and the pixels of each image were 4096×4096. The magnification of T4-15, T5-10 and T5-15 image was 2800 times, T4-6 image was 2000 times. – lines 114-117
Part Experimental results
- Again, please keep formatting and fonts, once used bolt, then italic and even regular type …..,
The formatting and fonts have been unified.
- Lines 131 and 133, image numbering should be separated by commas……,
The wrong writing has been corrected. – lines 183,185
- The whole article is written very chaotically. Different fonts are used, there are many mistakes in the text and it seems to me like a "quickly written" article. It lacks a comprehensive concept, it is very difficult to read, and the quality of the images and their descriptions are low quality and unreadable. For example pictures 11, 12, 13 and 14 description of the axis "x" or "y" - what does Gray mean?
In the compressed package we will upload the original image of the image. The Gray mean average gray value of micro area. – lines 139-152
- In the "Conclusion" section, specific values are missing, e.g. a number of insoluble phases are mentioned in the discussion, but the results of the experimental part should be summarized in the conclusion.
The specific values about insoluble phases and hardness have been summarized in the conclusion. – lines 454-456,470,471.

Reviewer 2 Report
This manuscript proposes analysis method of composition-microstructure-performance in an aluminum alloy. They adopted an averaged gray value of the images taken by optical microscopy to express microstructure. They demonstrated the correlation among composition, grain size, precipitate distribution and the averaged gray value in the manuscript. The information would be of interest to the readers of Materials. However, the method and results are not clearly described. My main concern is about the reproducibility and universality of the averaged gray value. It is true that the averaged gray value depends on the grain size and dislocation density as the authors mentioned in the manuscript but it also depends on the condition of observation including magnification and light intensity. In spite of that, I cannot find any detailed explanation about the observation and image processing conditions of optical microscopy to ensure the reproducibility and universality in Capt. 1. Overall, I would recommend major revision and re-reviewing before considering publication. One major comment is as follows:
1. Sec. 1.2., judging from line 280, optical microscopy was used for the gray value but any description of the method cannot be found in this section. The observation and image processing conditions for optical microscopy to ensure reproducibility and universality of averaged gray value should be explained.
Other minor comments are listed below:
2. Line 73, “tissue” should be “microstructure”.
3. Line 80, “steel plate” should be “aluminum plate”.
4. Sec. 1.2., the conditions of EBSD measurement should be explained.
5. Line 91, please add the resolution and magnification of the images for reproducibility.
6. Line 133, “(g)” should be “(j)”.
7. Second paragraph of Sec. 2.1.1., please include the information of the direction that the authors mention.
8. Figure 2, what does the “microstructure” mean? Is this IPF map or grain map? If it is IPF map, please add the orientation legend. Please add normal (ND), transverse (TD) and rolling direction (RD) in the figures of microstructure. Please add length scales.
9. Figure 2, which directions does “X0”, “Y0” and “Z0” mean? ND, TD and RD should be shown to show the texture. The color bars for IPFs should be enlarged to make it readable.
10. Line 172, “Fig. 3” should be “Fig. 4”.
11. Figure 4, please explain what (a) to (d) show in the figure caption.
12. Line 216 to 220, why the total area, quantity and average area are asymmetric?
13. Figure 7 and 8, please add ND, TD and RD.
Author Response
- Sec. 1.2., judging from line 280, optical microscopy was used for the gray value but any description of the method cannot be found in this section. The observation and image processing conditions for optical microscopy to ensure reproducibility and universality of averaged gray value should be explained.
The section 2.3 was added to describe the method of gray value acquisition. After adjusting the shooting conditions such as brightness and contrast, the metallographic microscope adopts automatic focus and continuous acquisition mode, which ensures the same parameters for the same batch of photos. Regarding reproducibility and universality, the same sample has been prepared and photographed for many times. There are differences between the gray values of the images obtained from different batches, but the relative change trend of the gray values of different parts of the entire surface is determined. The research of this paper is still in the stage of exploring the correlation between composition-microstructure-hardness, and the focus is to discuss the relationship between the distribution and trend of gray value and the other two sets of data. Finally, the experimental sequence of this paper is nondestructive experiment-destructive experiment, which basically guarantees that all data come from the same area.
Other minor comments are listed below:
- Line 73, “tissue” should be “microstructure”.
The wrong writing has been corrected. – line 96
- Line 80, “steel plate” should be “aluminum plate”.
The wrong writing has been corrected. – line 104
- Sec. 1.2., the conditions of EBSD measurement should be explained.
The conditions of EBSD measurement are explained in new section 2.6
- Line 91, please add the resolution and magnification of the images for reproducibility.
The number of images collected from four samples are 3362 (T4-6), 11508 (T4-15), 7056 (T5-10), 10668 (T5-15), and the pixels of each image were 4096×4096. The magnification of T4-15, T5-10 and T5-15 image was 2800 times, T4-6 image was 2000 times. – lines 114-117
- Line 133, “(g)” should be “(j)”.
The wrong writing has been corrected. “(g)” should be “(i)”. – line 183
- Second paragraph of Sec. 2.1.1., please include the information of the direction that the authors mention.
The direction information has been marked in the Figure.8.
- Figure 2, what does the “microstructure” mean? Is this IPF map or grain map? If it is IPF map, please add the orientation legend. Please add normal (ND), transverse (TD) and rolling direction (RD) in the figures of microstructure. Please add length scales.
The “microstructure” mean grain map. The normal (ND), transverse (TD) and rolling direction (RD) of microstructure have been add in the Figure 2. The length scales have been added in the Figure 2.
- Figure 2, which directions does “X0”, “Y0” and “Z0” mean? ND, TD and RD should be shown to show the texture. The color bars for IPFs should be enlarged to make it readable.
“X0”, “Y0” and “Z0” mean ND, TD and RD respectively. The color bars for IPFs have been enlarged to make it readable in Figure 2.
- Line 172, “Fig. 3” should be “Fig. 4”.
“Fig. 3” has been corrected “Fig. 4”. – line 242
- Figure 4, please explain what (a) to (d) show in the figure caption.
(a) Misorientation distribution of T4, (b) Misorientation distribution of T5, (c) Diameter distribution of T4, (d) Diameter distribution of T5.
- Line 216 to 220, why the total area, quantity and average area are asymmetric?
During the hot forming process of the rolled sheet, the surface layer and the middle layer are subjected to different forces and different holding times. As a result, the deformation degree of different part in rolled sheet is different, and the degree of fragmentation of the intermetallic compound is different. In addition, the holding time and element segregation will also affect the quantity and area of the insoluble phase.—line 280-285
- Figure 7 and 8, please add ND, TD and RD.
ND, TD and RD have been added in Figure 7 and 8.

Round 2
Reviewer 1 Report
After reading the new version of the article, it can be seen that the authors added the required information that was missing in the original version and reduced the professional level of the published results. A significant improvement can be seen in this direction. However, there is still a problem with the formatting of the document, specifically line 371, page 25: "The number of insoluble phases is also low in the position (image sequence number 200、400、600)where the gray value drops sharply based on Fig.11( c)." Again use commas to separate and add a space after the parenthesis.
Please check the formatting and editing of the text carefully.
Reviewer 2 Report
The manuscript has been improved well. I think current version is acceptable for publication.